# The Triple Adipose-Derived Stem Cell Exosome Technology as a Potential Tool for Treating Triple-Negative Breast Cancer

**DOI:** 10.3390/cells13070614

**Published:** 2024-04-02

**Authors:** Andrea Pagani, Dominik Duscher, Sebastian Geis, Silvan Klein, Leonard Knoedler, Adriana C. Panayi, Dmytro Oliinyk, Oliver Felthaus, Lukas Prantl

**Affiliations:** 1Department of Plastic, Hand and Reconstructive Surgery, University Hospital Regensburg, Franz-Josef-Strauß Allee 11, 93053 Regensburg, Germanysilvan.klein@klinik.uni-regensburg.de (S.K.); oliver.felthaus@ukr.de (O.F.); lukas.prantl@ukr.de (L.P.); 2Department of Plastic, Hand and Reconstructive Surgery, BG Klinik Ludwigshafen, University of Heidelberg, Ludwig-Guttmann-Straße 13, 67071 Ludwigshafen, Germany

**Keywords:** extracellular vesicles, exosomes, adipose-derived stem cells, tumor microenvironment, triple-negative breast cancer, T.A.E. technology

## Abstract

Background: Extracellular vesicles are pivotal mediators in intercellular communication, facilitating the exchange of biological information among healthy, pathological and tumor cells. Between the diverse subtypes of extracellular vesicles, exosomes have unique properties and clinical and therapeutical applications. Breast cancer ranks as one of the most prevalent malignancies across the globe. Both the tumor core and its surrounding microenvironment engage in a complex, orchestrated interaction that facilitates cancer’s growth and spread. Methods: The most significant PubMed literature about extracellular vesicles and Adipose-Derived Stem Cell Exosomes and breast cancer was selected in order to report their biological properties and potential applications, in particular in treating triple-negative breast cancer. Results: Adipose-Derived Stem Cell Exosomes represent a potential tool in targeting triple-negative breast cancer cells at three main levels: the tumor core, the tumor microenvironment and surrounding tissues, including metastases. Conclusions: The possibility of impacting triple-negative breast cancer cells with engineered Adipose-Derived Stem Cell Exosomes is real. The opportunity to translate our current in vitro analyses into a future in vivo scenario is even more challenging.

## 1. Introduction

Extracellular vesicles (EVs) are key in intercellular communication, facilitating the exchange of biological information among cells. Their intrinsic ability to transfer proteins, lipids and nucleic acids in small fragments influences several physiological and pathological functions inside of the body. In 2015, the European Cooperation in Science and Technology (COST) action initiative “*Microvesicles and Exosomes in Disease and Health*” (ME-HaD) confirmed that the current literature and knowledge about EVs is still poor and needs to be improved [1]. Current research projects use EVs to selectively target pathological cells while preserving healthy ones, thereby modulating the intercellular communication between them. Numerous studies about EVs have culminated in the creation of public online databases such as EVpedia, Vesiclepedia and Exocarta. These resources are instrumental in classifying different EVs, thereby facilitating a deeper understanding of their potential applications for various therapeutic needs [2,3,4].

Considering eukaryotic EVs, the most prominent subtypes are represented by microvesicles/ectosomes, apoptotic bodies and exosomes, which have gained particular importance for their unique properties and clinical and therapeutical applications. Exosomes are small vesicles (30–150 nm) that originate from the endosomal pathway and are released into the extracellular space upon the fusion of multivesicular bodies with the plasma membrane. Researchers are currently searching for valuable therapeutic options to engineer exosomes to control many diseases, aid in disease diagnosis or direct the delivery of a specific molecule to a desired target. For example, Ebrahimian et al. [5] showed how the local delivery of *Thymoquinone-Loaded Mesenchymal-Stem-Cell-Derived (MSC) Exosomes* is an efficient delivery system to bring the chemotherapy inside of the breast cancer core. In another study, Shojaei et al. [6] showed how *miR–218-enriched MSC* exosomes reduce the epithelial-to-mesenchymal transition (EMT) and angiogenesis, which are both key in the most malignant breast cancer, triple-negative breast cancer (TNBC).

TNBC represents the most aggressive and challenging breast cancer subtype worldwide. Growth and local invasion of the tumor core occurs at the expense of the surrounding tissues in a well-coordinated interplay [7]. The tumor microenvironment (TME) is key in cancer progression and represents a fundamental target for novel therapies, particularly against TNBC. While surgery, chemotherapy and radiotherapy remain the cornerstone treatments, research is actively pursuing novel therapeutic options for impacting TNBC at the core of the tumor (I), in its TME (II) and in surrounding tissues, including metastases (III). Because of the complexity of developing an in vivo technology that impacts the tumor at its three main levels (I, II and III), significant in vitro analyses have to be performed to exploit this challenging possibility.

In this review, the authors delve into the world of EVs, in particular, the biological properties and potential applications of Adipose-Derived Stem Cell Exosomes (ASC Exos). Upon detailing the biological attributes and functions of ASC Exos, we succinctly underscore their existing applications across various clinical scenarios, as well as their prospective efficacy against TNBC. Finally, we introduce the potential in vitro impact of engineered ASC Exos on TNBC cells to highlight the Triple ASC Exosomes [T.A.E.] Technology and show in which direction our research group is going in order to produce significant results in breast cancer research.

## 2. Extracellular Vesicles, Exosomes and Clinical Applications of ASC Exosomes

### 2.1. Extracellular Vesicles: Composition and Subtypes

Eukaryotic EVs are membrane-contained vesicles comprising different subtypes based on their content, size (from 10 μm to 30 nm) and membrane composition (e.g., cytoskeletal, cytosolic, heat shock and surface proteins). In addition to the traditional juxtacrine, autocrine and paracrine communication, EVs influence and communicate with the surrounding environment by transferring proteins, lipids and nucleic acids to the other cells [1]. The characterization of EVs is largely determined by their cytoplasmic content and membrane composition. However, the markers of EVs are shared with a multitude of vesicle types, and there is no unique surface protein that can definitively distinguish EVs, pinpoint their cellular origin or identify their distinctive characteristics [8,9,10].

The main EV subgroups are identified by their size and are represented by large oncosomes (1–10 μm) and exophers (3500–4000 nm), apoptotic bodies (1000–5000 nm) and migrasomes (500–3000 nm), small extracellular vesicles (30–1000 nm) and extracellular particles (<50 nm) (See Figure 1).

Whereas microvesicles are formed by outward budding of the plasma membrane, apoptotic bodies are released during programmed cell death, contain organelles and nuclear fragments and are involved in the clearance of dying cells by phagocytosis. Due to these characteristics, these vesicles are also called ectosomes. In contrast, exosomes are derived from the endosomal pathway and are released into the extracellular milieu following the fusion of multivesicular bodies (MVBs) with the plasma membrane (see Figure 2).

There are several isolation and characterization methods, such as ultracentrifugation, filtration, size exclusion or affinity chromatography, reactive ligands and antibody technology, that attempt to differentiate EVs subgroups. The paucity of exclusive markers for EVs complicates the distinction of the aforementioned subpopulations [1]. The current main proteins which mark EV subpopulations include tetraspanins (CD9, CD63, CD81 and CD82), the major histocompatibility complex (MHC), 14–3–3 proteins, tumor susceptibility gene 101 (Tsg101), heat-shock proteins and the Endosomal Sorting Complex Required for Transport 3 (ESCRT–3) [11,12,13]. Protein glycosylation patterns, such as N-glycans and N-acetyl lactosamine, along with lectin-binding profiles, for instance, galectin-3 binding, play significant roles in cellular sorting and homeostasis. However, these markers are not sufficient for identifying specific subgroups EVs.

EVs can rearrange the distribution of tetraspanins, ESCRT complexes and the BAR (Bin/Amphiphysin/Rvs)-domain-containing proteins, thus minimizing the cellular surface free energy, altering their structural curvature and promoting their release via cellular “budding” [14,15,16]. However, due to the small differences even in this last characteristic, the discrimination between different EV subgroups during their release remains challenging.

### 2.2. Exosomes: Composition, Biological Properties and ASC Exosomes

EVs can be classified into two main categories: large EVs, which are released through outward budding of the plasma membrane, and small EVs, with endosomal origin. The exosomal surface pattern composed of the Ras-related GTP binding protein B, tetraspanins, the soluble N-ethylamide-sensitive factor Activating protein Receptor (SNARE) complex proteins, Sytenin-1, Tsg101, the Apoptosis-Linked gene 2-Interacting protein X (ALIX), Syndecan-1, and ESCRT are correlated with their biogenesis and functional impact [17,18]. After being released, exosomes transport their bioactive content to the other cells through ligand/receptor interactions, membrane fusion and cargo internalization to influence the cellular microenvironment [11,19]. On the other hand, even host/local recipient cells can impact exosomes by secreting growth factors (GFs), cytokines and signaling molecules that can be taken up and change exosome content. The physiological condition of recipient cells, called a “*stress response*”, is activated in critical situations, and local microenvironment factors (e.g., pH or oxygen levels) and fine feedback mechanisms can impact the composition of exosomes, resulting in changes in their intrinsic nature and following functionality. Current research efforts, even of our research group, are aimed at engineering exosomes to manage a variety of diseases, enhance diagnostic measures and ensure targeted delivery to specific sites within the body.

Exosomes can be isolated from blood serum, plasma, saliva, urine, breast milk and amniotic and cerebrospinal fluid [20]. Among other tissue types, Adipose Tissue (AT) is a valuable source of high-quality exosomes, particularly those present inside the Stromal Vascular Fraction (SVF) [21]. In addition to immune cells, erythrocytes and endothelial cells, the SVF contains a particular subset of mesenchymal stem cells: Adipose-Derived Stem Cells (ASCs) [22,23]. Their particular mesenchymal-like morphology, the expression profile of CD34^+^, CD44^+^, CD31^−^ and CD45^−^, and their ability to produce osteoblasts, chondrocytes, myocytes and epithelial and neuronal cells allows ASCs to act as chemoattractant, angiogenic and prosurvival molecules [24]. The presence of ASCs gives to AT immunomodulatory, wound healing and regenerative properties [25,26]. All these functions are supported by additional mediators that live within the SVF and are released by ASC paracrine secretion, called ASC Exosomes. Even if both ASCs and ASC Exos co-exist within the SVF, they perform their functions via different mechanisms. In contrast to ASCs, ASC Exos orchestrate tissue regeneration, immunological functions and tissue and organ homeostasis by acting as versatile vesicles and avoiding cell-therapy-associated problems such as cell survival, unfavorable differentiation, senescence-induced instability and immune rejection [21].

Because of these specific advantages and the possibility of harvesting them easily, ASC Exos are gaining particular attention. By inducing cell migration and proliferation, ASC Exos enhance tissue regeneration, impacting skin aging, wound healing, scar quality, flap vitality and fat grafting [21,27,28]. Because of their immunomodulatory features, ASC Exos have potential anti-inflammatory properties even against osteoarthritis [29]. In a recent study, Fang et al. successfully treated CCl4-induced acute liver injury with MSC Exos after engineering them with Quercetin to enhance their therapeutic efficacy and with Vitamin A to selectively target the liver. This significantly reduced the rapid senescence-like response induced by acute liver injury [30]. In another paper, Qian et al. demonstrated how hypoxic ASC Exos can attenuate colitis by regulating macrophage polarization [31]. Even in the neurological field, ASC Exos showed promising in vitro results by ameliorating a phenotype of Huntington’s disease [32]. Altogether, there are several ongoing studies that seek to prove the efficacy of exosomes in treating several clinical conditions. Despite their in vitro effectiveness and consequent potential clinical applications, ASC Exos need more clinical trials to establish their efficacy and safety for clinical use. By following this ethical concept, our research group is trying to define an exosome technology able to modulate in vitro the host and recipient cells of TNBC before translating the laboratory results into a clinical reality.

## 3. Breast Cancer as Triple Entity: Classification System and General Therapeutical Options

According to WHO data and Globocan 2020, breast cancer represents the most commonly diagnosed cancer worldwide, with an estimated 2.3 million new cases in 2020 [33]. Breast cancer has evolved in our understanding from a solitary oncological mass to a dynamic constellation involving a central core, a responsive local microenvironment and an interplay with adjacent healthy tissues. The TME, once relegated to a non-active role, is now recognized as a critical influencer within the oncological landscape, facilitating tumor genesis and promoting the progression of cancer [7]. As cancer invasion is dictated by the mutual interactions between the tumor core (I), its microenvironment (II) and the surrounding tissues (III), the most innovative therapeutic strategies should act on these three elements.

The interplay between a cancerous mass and its local TME is orchestrated through a symphony of direct cell-to-cell interactions, paracrine signaling and the exchange of vesicles. Among the myriad of signaling entities, tumor-derived exosomes are prominent players in this intricate dialogue, exerting a profound influence on the processes of cancer invasion, metastasis and the development of chemoresistance [34]. The possibility of taking advantage of the pro-oncogenic role of tumor exosomes and developing an ASC-exosome-based therapy against breast cancer is real. However, novel ideas and technologies have to be tested first in vitro and then in vivo.

### 3.1. Breast Cancer Classification Systems: Tumor Size, Local Receptors and Immunohistochemistry

As with all solid tumors, the traditional TMN classification for breast cancer is current, valid and effective for staging a tumor [Figure 3].

From the perspective of anatomical pathology, breast cancer is categorized according to its histological type, immunohistochemical characteristics or receptor profile. The predominant histological types are ductal and lobular carcinoma. These are further classified as either in situ or invasive, contingent upon whether the tumor breaches the basement membrane. Among the less common varieties are mucinous carcinoma, primary breast lymphoma and neuroendocrine, medullary, and inflammatory carcinoma [35]. From the immunohistochemical side, Luminal-A (40%) and -B (20%), Normal-like (2–8%) and Her2-enriched (10–15%) tumors represent the most prominent subtypes. TNBC represents the most dangerous breast malignancy (15–20%); it does not possess receptors for estrogen, progesterone or Her2–Neu and also usually has a high growth rate (Ki67) (see Figure 4).

Despite receiving standard frontline therapy, TNBC often exhibits recurrence or metastasis within the first 3 years post-surgery [36]. In addition, Goh et al. [37] recently reported that the 5-year survival rate for TNBC patients is 77%, compared to 93% for other breast cancer subtypes. Despite it being traditionally considered a “basal-like” breast cancer, TNBC is currently divided into six different subtypes: (a; b) basal-like 1 and 2, (c) immunomodulatory, (d) luminal androgen receptor, (e; f) mesenchymal and mesenchymal-stem-like. All these categories present different responses to standard chemotherapy [38].

### 3.2. Basic Knowledge and Novel Therapeutical Options for Breast Cancer Therapy

As widely highlighted by Brett et al. (2022), traditional surgery, chemotherapy and radiotherapy represent the main traditional options for breast cancer. The therapeutical strategy is based on the severity, the clinical condition of the patient and their response to treatment. Whereas surgery is indicated for breast tumors in stages I, II or III, radiotherapy is performed for the higher stages and shrinks the tumor core and the surrounding tissues in a non-specific way [7].

The standard chemotherapy regimen for breast cancer includes Anthracyclines (Doxorubicin and Epirubicin) and Taxanes (Docetaxel and Paclitaxel). Tamoxifen, a Selective Estrogen Receptor Modulator (SERM), blocks estrogen effects in estrogen-sensitive women, contributing to the reduction in tumor development. Conversely, postmenopausal women may benefit from the androgen and anabolic steroid Fluoxymesterone, as well as the aromatase inhibitors Exemestane, Anastrozole and Letrozole. Other valuable options are represented by the monoclonal antibodies Trastuzumab, Lapatinib and Bevacizumab, most of the time in combination with Paclitaxel [7]. In addition to these consolidated therapeutical options, blocking or knocking down the CCL5/CCR5 axis is also a potential way to control breast cancer progression and invasion, even metastasis formation. The possibility of targeting the CCL5/CCR5 axis and inducing an anti-tumor environment is real and will be discussed in the last section of this paper. However, despite these innovative tools, prognosis is strongly influenced by the clinical stage at which the cancer is diagnosed.

## 4. TNBC and Tumor Exosomes: Mutual Friends or Potential Enemies?

The mutual interplay between tumor cells and their tumor-derived exosomes has long been a subject of debate. Initially, it appeared that exosomes primarily fostered tumor growth by inducing proliferation, EMT transition, immunological switch, neo-angiogenesis and metastasis. However, recent research showed that exosomes can shift from being allies to becoming formidable adversaries of the tumor mass. Goh et al. [37] widely discussed this topic, wondering if exosomes were a humble “garbage disposal” or brave “Trojan horses” in TNBC.

### 4.1. TNBC and Tumor Exosomes: Mutual Friends…

First of all, there is extensive literature that shows how breast cancer patients present a higher concentration of exosomes in their plasma compared to healthy controls. The oncological upregulation of the p-53 gene product “TSAP 6” in women with breast cancer, for example, enhances the concentration of exosomes in their blood plasma [39,40,41]. To support this study, Kavanagh et al. [42] showed that treatment with Paclitaxel results in the development of chemoresistant therapy-induced senescent (TIS Cal51) cells and induces higher exosome production than non-senescent control cells, which can then be released into the blood stream.

Second, tumor exosomes can be modulated by the TME and from their cells of origin. O’Brien et al. [43] demonstrated that TNBC Exos transfer phenotypic traits to secondary cells representing their cells of origin. By comparing the effects of tumor exosomes derived from the TNBC cell line Hs578T and its more invasive Hs578T**(i)8** variant, the authors showed that Hs5787T**(i)8**-derived Exos increase the proliferation, migration and invasiveness of the SKBR3, MDA-MB-231 and HCC1954 cell lines in a most aggressive way when compared with Hs578T-derived Exos. In addition, Hs5787T**(i)8** Exos were able to increase sensitivity to anoikis on recipient cells, reflecting the enhanced sensitivity of Hs578Ts**(i)8** to anoikis. Finally, Hs5787T**(i)8** Exos stimulated the formation of significantly more endothelial tubules, influencing even the vascular profile and, by proxy, tumor angiogenesis, of the recipient site when compared with the less invasive Hs578T cell line. All these statements prove that the oncological origin of tumor exosomes gives them a pro-oncogenic role within the tumor and its microenvironment.

Lastly, there is also a well-documented relationship between autophagy, exosomes and tumorigenesis [44,45,46,47,48,49,50,51]. In their comprehensive work, Li W et al. explained that exosome-mediated autophagy can upregulate cancer development, increasing proliferation and invasion and inducing immune evasion [52]. In particular tumor contexts, exosome release and autophagy are strongly activated, playing a coordinated role in tumor promotion [53,54]. The main oncological stressors that activate both the autophagic and exosomal pathways are unfolding protein response and endoplasmic reticulum stress. Because there is no current therapy that can stop these stressors from happening, a large number of clinical studies are developing autophagy inhibitors, particularly the recently FDA-approved drug Chloroquine/Hydroxychloroquine, in association with anti-cancer drugs, to sensitize chemoresistant cells to cancer treatment [55]. Despite the great interest in this topic, a better understanding of the biological dialogue between exosomes and autophagy is needed.

Not only the exosomes themselves but also their intravesicular components play an important oncogenic role. Bobrie et al. [56] and Yang et al. [57], for example, showed how the protein Rab27a or exosomal miR–223 facilitate exosome secretion and promote TNBC progression and invasiveness. Last but not least, tumor exosomes play a role in drug resistance. There is brilliant literature from Boelens et al. [58] and Diluvio et al. [59] that explains how stromal cell exosomes can mediate drug resistance in TNBC cells by modulating the NOTCH3 pathway. A significant portion of the scientific literature perceives exosomes as genuine allies of the tumor, capable of fostering a pro-tumor microenvironment that facilitates cancer progression, drug resistance and both local and distant metastasis.

If considering all these research papers, the pro-oncogenic role of tumor exosomes is more than supposed. Despite the great power of the literature behind their pro-oncogenic role, there has been a significant growth in novel papers that show that these abilities can be reversed and used as potential weapons against cancer.

### 4.2. TNBC and Tumor Exosomes: … or Potential Enemies

Considering the exosome-mediated role in tumor progression and invasion, the possibility to reverse exosome function has great potential. The first significant study about the potential therapeutical impact of EVs, and specifically of tumor EVs, towards breast cancer was performed by the group of Camargo. By using EVs from NFAT3-expressing cells, the authors were able to impede tumor growth and metastasis in a TNBC mice model [60]. This study opened a novel research pathway on this topic that revealed a promising tumor-suppressor role.

The exosomal pathway presents opportunities for intervention at multiple biological stages, encompassing biogenesis, release and uptake. Exosomes can be transformed into delivery systems to carry chemotherapeutic drugs or nucleic acids [37]. Moreover, it is possible to selectively silence specific exosomal cargoes to impact the cellular microenvironment. In the aforementioned studies of O’Brien et al., Bobrie et al. and Colletti et al., after showing the pro-oncogenic role of tumor exosomes, the authors explained in the following sections how this ability can be reversed, making exosomes a fatal enemy for breast cancer. One example is represented by Rab27a, which showed on one side an exosome-mediated pro-oncogenic role towards a mammary tumor in a mouse model; however, in the second part of the paper, Rab27a silencing resulted in reduced local growth and metastasis [56]. Even in the abovementioned study of O’Brien, the authors were able to show in the second part of the paper how the exosome-mediated transfection of miR–134 into Hs578Ts(i)8 cells was associated with a lower expression of STAT5B, Bcl–2 and HSP90, with consequent reduced tumor cell proliferation [43].

In a different paper by Li Y et al. [61], the authors modified the TME of TNBC by engineering exosomes with miR–770. The TNBC cells became more sensitive to Doxorubicin because of a synergistic action between miR–770 and Doxorubicin in therapeutic-induced apoptosis. Doxorubicin can also be loaded inside of exosomes in order to transport chemotherapeutic drugs with low toxicity and immunogenicity. Gomari et al. [62] reported how Doxorubicin-loaded exosomes were able to target Her2^+^ cells and showed a reduction in the tumor growth rate and reduced adverse effects in a murine breast cancer model.

The prospect of the surface modification of exosomes using specific peptides or tissue-specific antibodies to fine tune their homing capabilities is also a highly intriguing avenue of research. In the aforementioned study by Gomari, it was reported that exosomes could be engineered to target Her2^+^ tumor cells by incorporating pLEX-LAMP DARPin, thereby enabling DARPin expression on the exosome’s surface.

The last but not least mention is a study by Huang et al. [63], who engineered exosomes as an in situ dendritic cell (DC) primed vaccine in order to boost anti-tumor immunity in breast cancer. After producing an in situ DC vaccine (HELA–Exos) by loading the immunogenic cell death inducers human neutrophil elastase and Hiltonol into alpha-lactalbumin (α–LA)-loaded breast cancer exosomes, the authors were able to introduce a potential target therapy for breast cancer (see Table 1).

As the research article of Goh et al. reported, which gave us the inspiration for starting this journey, there is a need to included exosomes in TNBC research and, in particular, ASC exosomes. The possibility to use them as Trojan horses for TNBC therapy is real and is the focus of our first in vitro research study.

## 5. The Triple ASC Exosome [T.A.E.] Technology as an In Vitro Preconditioning Tool for TNBC Cells

In the perpetual fight against breast cancer, novel technologies try to offer hope for effective treatments and improved patient outcomes. The lack of ER, PR and Her2 receptors in TNBC means neo-adjuvant chemotherapy and/or radiotherapy first and traditional surgery after. Because of our clinical duty as plastic surgeons, the possibility of isolating, harvesting and engineering ASC Exos from AT is real but challenging. Hence, we are testing a novel Triple ASC Exosome [T.A.E.] Technology able to impact TNBC cells in vitro to find a proper ASC Exos combination with particular molecules that could act as an integrated and supportive tool for use in traditional breast cancer therapies. 

As mentioned before, their intrinsic low toxicity, low immunogenicity, high-flexibility engineering, inherent targeting and interplay with recipient cells make ASC Exos ideal drug carriers for breast cancer therapy. Jung et al. [65] developed an exosome platform to target regions of tumor hypoxia. After isolating four types of exosomes from MDA-MB-231 human breast cancer cells, the authors engineered them to create supermagnetic iron oxide (SPIO) nanoparticles with the anti-tumoral PARP inhibitor Olaparib to increase apoptosis and reduce tumor growth in vivo. By being inspired from this study but also from all the abovementioned literature, we decided to follow the research works of Cortes et al. (2015), Hu et al. (2021), Pervaiz et al. (2021) and Jiao et al. (2018) and considered the possibility to engineer ASCs Exos with Pembrolizumab, the Fibroblast Activation Protein-alpha (FAP) and Maraviroc (MVC) [63,66,67,68].

At the heart of the T.A.E. Technology lies the in vitro transformative power of ASCs and their related ASC Exosomes. Central to the T.A.E. Technology should be its potential ability to locally target TNBC cells within the tumor core, thereby bypassing systemic barriers and minimizing off-target effects. Engineered exosomes can deliver anti-cancer drugs, RNA interference molecules or immune-modulating molecules specifically to TNBC cells, inducing apoptosis or inhibiting proliferation to minimize off-target effects and enhance therapeutic efficacy. As mentioned in the previous section, tumor exosomes bestow several advantages upon the tumor core, including the activation of stromal components, initiation of the angiogenic switch, enhancement of vascular permeability, facilitation of the epithelial-to-mesenchymal transition and the establishment of a pre-metastatic niche [69].


*#1: Pembrolizumab–ASC Exosomes*


As an example of one of the possibilities of targeting the tumor core, we are currently testing in vitro with TNBC cells the first of the three T.A.E. components by engineering ASC exosomes with Pembrolizumab to deliver it within the tumor core cells. In a phase 3 trial of Cortes et al. [66], the addition of Pembrolizumab to chemotherapy resulted in longer overall survival than chemotherapy alone among TNBC patients whose tumors expressed programmed death ligand-1 (PD–L1) with a combined positive score (CPS) of 10 or more. 


*#2: FAP–ASC Exosomes*


Beyond the tumor cells themselves, the TME plays a key role in TNBC progression and treatment resistance. The second T.A.E. component should therefore impact this intricate milieu, leveraging ASC exosomes to modulate the TME in favor of anti-tumor immune responses via cytotoxic T cells or Natural Killer (NK) cells and inhibit pro-tumorigenic signals.

Before the beginning of our study, Hu et al. [63] demonstrated that engineered exosomes can be used as vaccines to reprogram the immune response of the TME. By producing Fibroblast Activation Protein-alpha (FAP) gene-engineered tumor-cell-derived exosome-like nanovesicles (eNVs–FAP) as a tumor vaccine, the authors were able to inhibit tumor growth, induce cytotoxic T lymphocyte (CTL) immunity against colon, melanoma, lung and breast cancer models, and reprogram the TME. Because of its activity, eNVs–FAP-engineered ASC Exos represent a special candidate as a tumor vaccine to be used for our second T.A.E. component to locally target the tumor parenchyma cells and the surrounding cellular environment.


*#3: MVC–ASC Exosomes*


The third T.A.E. component should be developed for the cellular component surrounding tissues or breast cancer metastasis. As exosomes derived from metastatic breast cancers have natural organotropism to the lung and brain, therapeutic drugs can be encapsulated with exosomes as a third component to address the different metastatic foci. Huang et al. [64], for example, loaded a nanomaterial for photothermal therapy, named as gold nanorods, into metastatic lung-cancer-cell-derived exosomes, and it exhibited better therapeutic effects on lung metastasis. Another therapeutical option is represented by Maraviroc (MVC), the non-peptidic antiretroviral CCR5 blocker that is used for HIV patients. Pervaiz et al. [67] analyzed the relationship between MCV and breast cancer. In one of their preclinical studies, the group demonstrated that blocking CCR5 reduces breast cancer proliferation, colony formation and metastasis. Furthermore, MVC inhibits bone metastasis in implanted MDA-MB-231 breast cancer cells in rats. Even Jiao et al. [68] inhibited TNBC cell proliferation and migration via IL–6 and CCL5 by exploiting MVC and Tocilizumab. Hence, an MVC–ASC–Exos complementary therapy inhibiting these mechanisms could be a valuable weapon against TNBC cells in vitro (see Figure 5).

At present, T.A.E. Technology is undergoing laboratory development with the potential to address and support traditional chemotherapeutical treatments against TNBC, with the final prospect of a future where TNBC can be controlled and effectively treated with a combination treatment. As research progresses and clinical trials advance, the transformative impact of “T.A.E. preconditioning” in TNBC treatment is poised to be realized, ushering in a new era of hope and healing in the fight against cancer.

## 6. Conclusions

TNBC remains one of the most common and dangerous malignancies worldwide. Due to the lack of ER, PR and Her2 expression, TNBC defies conventional therapies, necessitating novel therapeutical approaches, in particular for the pre-surgical treatment of the tumor mass. The possibility of engineering ASC exosomes and using them as in vitro weapons against TNBC cells is challenging. By targeting not only the tumor core cells but also its microenvironment cellular compartment and surrounding cells, including metastases, researchers have worked to analyze first of all the in vitro impact of engineered ASC exosomes against TNBC cells and then to develop a supportive T.A.E. Technology able to positively impact TNBC patients by performing a preconditioning treatment via engineered ASC exosomes. The possibility to exploit the transformative impact of ASC Exos within the T.A.E. technology is real, highlighting a new era for supportive measures against breast cancer.

## Figures and Tables

**Figure 1 cells-13-00614-f001:**
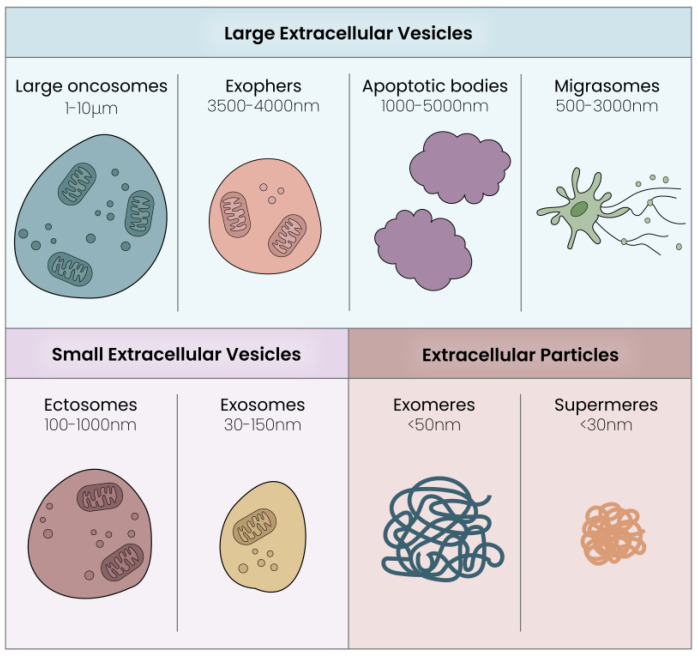
Extracellular vesicles: subtypes, dimensions and graphical concept. The different EV subtypes include large EVs (e.g., oncosomes, exophers, apoptotic bodies and migrasomes), small EVs (e.g., ectosomes and exosomes) and extracellular particles (EPs) including exomeres and supermeres.

**Figure 2 cells-13-00614-f002:**
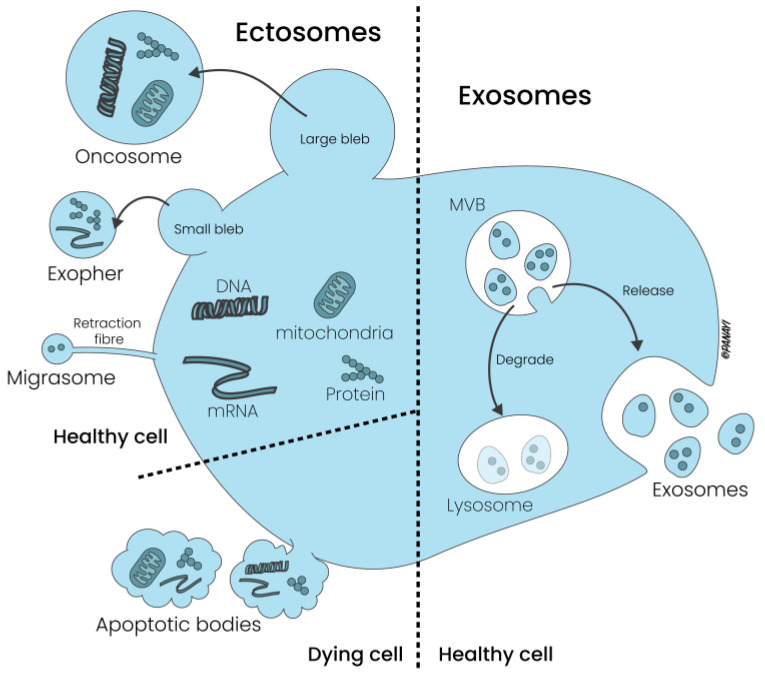
Generation and release of ectosomes and exosomes from the cellular membrane. While ectosomes are generated by a series of rapid steps, exosomes are released into the extracellular milieu following the fusion of multivesicular bodies with the plasma membrane.

**Figure 3 cells-13-00614-f003:**
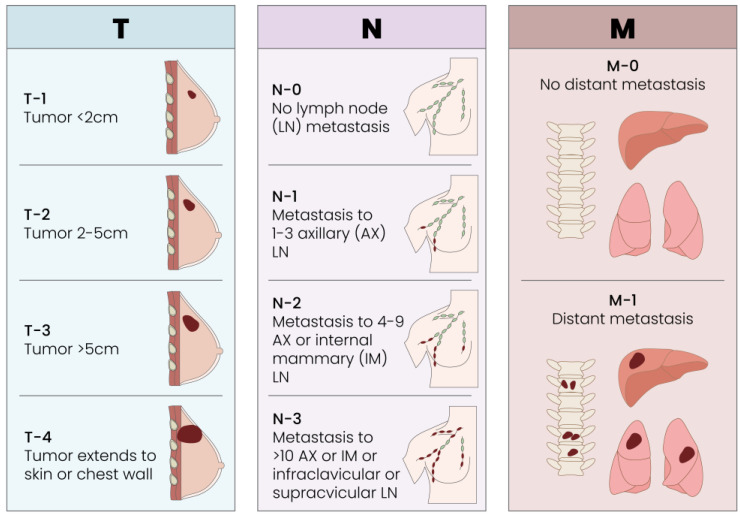
Breast cancer TNM classification. T represents the size of the tumor and the spread of cancer into nearby tissue, N describes the oncologic spread to nearby lymph nodes, and M describes metastasis.

**Figure 4 cells-13-00614-f004:**
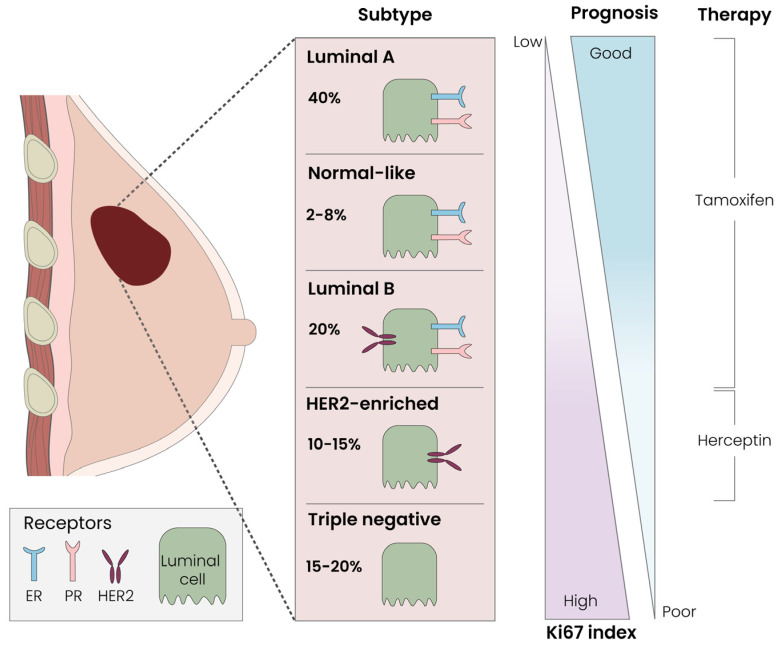
Breast cancer subtypes, prognosis and therapy. Every breast cancer subtype has a particular Ki67 index, related prognosis and therapeutical approach.

**Figure 5 cells-13-00614-f005:**
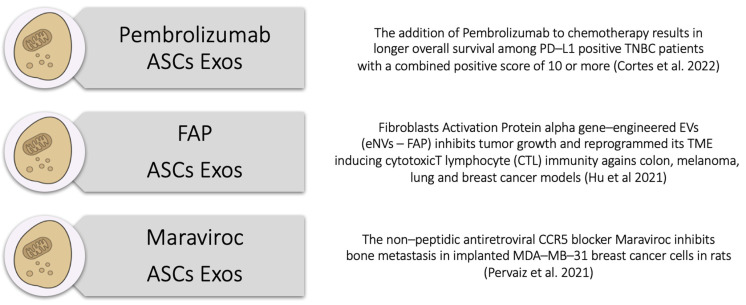
The T.A.E. Technology. Schematic table of the three different components of the T.A.E. Technology [63,66,67].

**Table 1 cells-13-00614-t001:** TNBC and ASC exosomes. Here we compare the different PubMed studies which report the relationship between TNBC and ASC exosomes as mutual friends (left side) or dangerous enemies (right side). The papers of O’Brien et al. (2013) [43], Colletti et al. (2021) [54] and Bobrie et al. (2012) [56] present both aspects of the interplay between TNBC and ASC Exos.

TNBC & ASCs EXOSOMES	
Mutual Friends…. …or Enemies	
Yu et al. [39] TSAP–6 upregulation enhances exos concentrations in blood plasma of breast cancer women	Camargo et al. [60] NFAT–3 expressing cells derived EVs impede tumor growth and metastasis in a TNBC mice model
Galindo-Hernandez et al. [41] Elevated concentrations of microvesicles from peripheral blood in breast cancer patients	Gomari et al. [62] pLEX–LAMP DARPin–loaded Exos can enable DARPin expression on their surface and effectively address Her–2^+^ tumor cells
Kavanagh et al. [42] TIS Cal 51 chemoresistant cells produce more exosomes after Paclitaxel treatment	Gomari et al. [62] Doxorubicin–loaded Exos target Her–2^+^ cells with reduction of tumor growth and reduced adverse effects in a murine breast cancer model
O’ Brien et al. [43] Malignant Hs5787T(i)8–derived Exos increase proliferation, migration, invasiveness, endothelial tubules formation and angiogenesis of recipient cells (TME)	O’ Brien et al. [43] Exosomes–mediated transfection of miR–134 into Hs578Ts(i)8 cells is associated with reduced tumor cells proliferation
Li W et al. [52] Exosomes–mediated autophagy upregulated cancer development impacting proliferation, invasion and immune evasion	Li Y et al. [61] Engineered miR–770 Exos are more sensitive to Doxorubicin in therapeutic–induced apoptosis
Colletti et al. [54] An oncological environment activates exosomes release and authophagy which support tumor growth	Colletti et al. [54] An oncological environment activates exosomes release and authophagy which support tumor suppression
Bobrie et al. [56]–Yang et al. [57] Exosomal Rab27a and miR–223 supports exosomes secretion and TNBC progression and invasiveness	Bobrie et al. [56] Silencing the Exosomal Rab27a reduces local growth and tumor metastasis
Boelens et al. [58]–Diluvio et al. [59] Stromal cells–derived Exos mediate drug resistance in TNBC cells by modulating the NOTCH–3 pathway	Huang et al. [64] HELA–Exos is able to boost antitumor immunity in breast cancer

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
