# Peer review of "The Triple Adipose-Derived Stem Cell Exosome Technology as a Potential Tool for Treating Triple-Negative Breast Cancer"

_cells, 2024, doi:10.3390/cells13070614_

Round 1

Reviewer 1 Report

Comments and Suggestions for Authors

Very interesting prospective review

p9 paragraph 4.2

I think this reference should be included because it's one of the first paper showing that tumor exosomes (from luminal breast cancer cells) can impede tumor growth and metastases arising in a TNBC mice model.

Camargo, L. C. B. de et al. Extracellular vesicles produced by NFAT3-expressing cells hinder tumor growth and metastatic dissemination. Sci Rep 10, 8964 (2020).

Author Response

  1. Camargo, L. C. B. de et al.Extracellular vesicles produced by NFAT3-expressing cells hinder tumor growth and metastatic dissemination. Sci Rep 10, 8964 (2020).I think this reference should be included because it's one of the first paper showing that tumor exosomes (from luminal breast cancer cells) can impede tumor growth and metastases arising in a TNBC mice model.

Thank you very much for your suggestion. After updating the section with the reference of Camargo et al. “The first significant study about the potential impact of extracellular vesicles, and specifically of tumor vesicles, towards breast cancer was performed by the group of Camargo. By using EVs from NFAT3–expressing cells, authors were able to impede tumor growth and metastasis in a TNBC mice model”, we additionally reviewed the entire manuscript and, specifically, the Section 4.2 in order to make it as readable as possible.

Reviewer 2 Report

Comments and Suggestions for Authors

The manuscript by Pagani A. et al. explores the potential utility of Adipose-Derived Stem Cells Exosomes in targeting Triple-Negative Breast Cancer (TNBC) and its microenvironment, including metastatic tissues. The authors propose the development of Triple-Adipose-Derived Stem Cells Exosomes T.A.E., to precondition TNBC for traditional oncological treatments.

Overall, the manuscript presents an interesting topic related to TNBC treatment, though some issues have to be addressed.

First of all, when introducing EV, author should specify they are going to describe only eukaryotic vesicles.

Page 4 lines 136-137, how does the recipient cell influence exosome content. Please clarify and cite references.

Page 5, lines 159-162, lipoaspiration is actually not a very simple way, it is simple for plastic surgeons.

The authors should provide a more comprehensive explanation of how this treatment option could be developed and implemented in the clinical environments of oncology and plastic surgery.

Page 6, the paragraph on tumor staging is not well-focused on EVs.

Page 6 line 165, what were the results of the cited study [30]?

Page 6, line 182, “these three entities”, which ones?

Page 8, Subtitles would be more effective without being formulated as questions.

Author Response

  1. When introducing EV, author should specify they are going to describe only eukaryotic vesicles.

    In the Introduction and in the second section of the manuscript, we specified the eukaryotic nature of EVs.

  1. Page 4 lines 136-137, how does the recipient cell influence exosome content. Please clarify and cite references.

    Thank you for helping us to better this unclear section of our manuscript. We updated the paper considering the impact of recipient cells on local exosomes.

  2. Page 5, lines 159-162, lipoaspiration is actually not a very simple way, it is simple for plastic surgeons.

    By considering your suggestion, another major revision (Reviewer #4) and in order to address this paper even to other medical fields, we formulated a new updated paper exploiting the in–vitro potential application of our technology. We do not mention lipoaspiration in our new paper anymore. A manuscript with the clinical application of the technology will be written as soon as the first significant in–vitro results will be achieved.

  1. The authors should provide a more comprehensive explanation of how this treatment option could be developed and implemented in the clinical environments of oncology and plastic surgery.

    In order to present our current understanding about the potential impact of engineered ASCs Exosomes towards TNBC cells, we formulated a new updated work highlighting the potential in–vitro application of our technology. In the new version, there is no real statement about a clinical application of ASCs Exosomes towards TNBC. About this, we will prepare a new manuscript when the first in vitro results will be achieved.

  1. Page 6, the paragraph on tumor staging is not well-focused on EVs.

    The paragraph should not have necessarily a relationship with EVs. It serves as first introduction for the TNBC in order to consider its traditional classification system and its current therapeutical options.

  1. Page 6 line 165, what were the results of the cited study [30]?

    We updated the manuscript including the results with Quercetin– and Vitamin A–loaded Exos of Fang et al. (2021).
  1. Page 6, line 182, “these three entities”, which ones?

    We explained better the concept with the following sentence: “Exactly as cancer invasion is dictated by mutual interaction between the tumor core (I), its microenvironment (II) and the surrounding tissues (III), the most innovative therapeutic strategies should act on these three elements. “
  1. Page 8, Subtitles would be more effective without being formulated as questions.

    Thank you for the suggestion. We updated the subtitles with the following statements: “Mutual friends… or potential enemies.”

Reviewer 3 Report

Comments and Suggestions for Authors

Review Report manuscript ID cells-2926774

General Considerations: The authors have explored a new sector in cancer research, specifically regarding breast tumors. They have focused on the role of exosomes, highlighting their unique properties and clinical and therapeutic applications. The manuscript is well-organized and elaborately described in various sections, and it could significantly contribute to the emerging field of extracellular vesicles in cancer research.

Major Revision:

·      There is no necessary explanation of the histological properties of breast cancer subtypes. This section could be resumed and summarized in a shorter paragraph.

·      The review article does not explore the crosstalk between exosomal and autophagic pathways. However, recent studies have shown that exosomes can regulate the intracellular autophagic process, which, in turn, affects circulating exosomes. Therefore, adding a new paragraph to enrich the review would be better. In particular, section 4, where the authors also show the links between exosomes and senescence in TNBC, could be enriched with some papers showing the roles of autophagy and the exosome in this type of tumor.

Minor revisions

·      Please add a graphical abstract to your review to recap the principal messages of the manuscript and focus readers' attention on this paper.

Comments on the Quality of English Language

 Minor editing of the English language is required.

Author Response

Major Revision:

  1. There is no necessary explanation of the histological properties of breast cancer subtypes. This section could be resumed and summarized in a shorter paragraph.

    Thank you for the useful suggestion. We summarized the section in a shorter paragraph.

  1. The review article does not explore the crosstalk between exosomal and autophagic pathways. However, recent studies have shown that exosomes can regulate the intracellular autophagic process, which, in turn, affects circulating exosomes. Therefore, adding a new paragraph to enrich the review would be better. In particular, section 4, where the authors also show the links between exosomes and senescence in TNBC, could be enriched with some papers showing the roles of autophagy and the exosome in this type of tumor.

    This suggestion helped us to significantly better the 4th section of the manuscript. We added the following part within the paper by mainly considering and citing the valuable works of Li et al. (2018), Colletti et al. (2020) and Singh et al (2018).

Minor revisions

  1. Please add a graphical abstract to your review to recap the principal messages of the manuscript and focus readers' attention on this paper.

    We provided a graphical abstract of the manuscript and other new tables for the manuscript.

Reviewer 4 Report

Comments and Suggestions for Authors

This manuscript addresses an important topic concerning the role of extracellular vesicles in intercellular communication, facilitating the exchange of biological information among healthy and tumor cells. This article has certain research value, but the research content is not sufficient and there is a lack of relevant mechanism research. The Reviewer has the following concerns regarding the clarity, coherence, and scientific rigor of the manuscript:

1. To ensure a proper assessment of the biological function of exosomes, the authors should provide higher resolution images of all schematic diagramsAt least two new summative figures are needed (Two figures and one table say nothing about the strength of this review). The authors should draw upon the recent articles that they review to suggest new research directions, to strengthen support for existing theories and/or identify patterns among existing research studies.

2. There are multiple instances where the review lacks clarity and coherence. There are sentences randomly lifted from somewhere that doesn't fit into the context. The primary goal of this study was to determine whether developing a supportive Triple–Adipose-derived stem cell exosomes [T.A.E.] technology against breast cancer. It doesn't make sense, as this was not an experimental study, but a review article.

3. The review does not provide a clear opinion on topics, there is a very generic introduction to the topic but not thorough analysis. A detailed table highlighting the key studies or experimental evidence supporting the association between exosomes and breast cancer is missing.

4. The paper in its present form fails to interpret the data in the context of what is known in the field: it sounds somehow redundant, without placing them in the proper scientific context.

5. The paper is mainly descriptive and focused on its (not fully supported) conclusions.

Comments on the Quality of English Language

English language (syntax, grammar, correct choice of words, correct use of adjectives and adverbs) needs significant editing throughout the text and figures. Professional assistance must be sought.

Author Response

  1. To ensure a proper assessment of the biological function of exosomes, the authors should provide higher resolution images of all schematic diagrams. At least two new summative figures are needed (Two figures and one table say nothing about the strength of this review). The authors should draw upon the recent articles that they review to suggest new research directions, to strengthen support for existing theories and/or identify patterns among existing research studies.

    High resolution figures are provided in the manuscript. Unfortunately, the MDPI-Homepage downgrades the quality for the reviewers during the review process. As suggested, we added a graphical abstract (GA) at the beginning of our manuscript and two further tables (Table 3 and 4) for the fourth and fifth sections.

  1. There are multiple instances where the review lacks clarity and coherence. There are sentences randomly lifted from somewhere that doesn't fit into the context. The primary goal of this study was to determine whether developing a supportive Triple–Adipose-derived stem cell exosomes [T.A.E.] technology against breast cancer. It doesn't make sense, as this was not an experimental study, but a review article.

    In order to propose a valid, descriptive but more structured Review on Breast Cancer Therapy Research via ASCs Exos, we modified the old version of the manuscript in every single section, starting from the title. This to propose a more schematic and clear version. Given the little quantity of research article and, consequently, of solid advices about this topic, we propose now a “potential in–vitro technology” that can impact TNBC cells in order to start our laboratory analyses about this topic but even motivate other readers to start doing research in this field.

  1. The review does not provide a clear opinion on topics, there is a very generic introduction to the topic but not thorough analysis. A detailed table highlighting the key studies or experimental evidence supporting the association between exosomes and breast cancer is missing.

    Thank you for your suggestion, which helped to increase the quality of one section of the manuscript. As you explained, a systematic and clear table showing the relationship between exosomes and breast cancer was missing. Hence, we added a new table at the end of the fourth section in order to compare the dual role of exosomes towards breast cancer. In addition, the fourth section of the manuscript was update in order to analyze more precisely the current knowledge and interplay between exosomes and breast cancer.

  1. The paper in its present form fails to interpret the data in the context of what is known in the field: it sounds somehow redundant, without placing them in the proper scientific context.

    By considering the suggestions of the other authors and your other suggestions (Point 2 and 3), we updated a new version of the manuscript, eliminating redundant sentences and placing new sections in the proper context. In this new manuscript, the authors consider the TAE Technology as an in–vitro tool to address TNBC Cells via engineered ASCs Exosomes. This to highlight our current in vitro–analyses with Pembrolizumab / FAP / MVC – loaded Exosomes. When the laboratory analyses will be significant, we will prepare a new manuscript in order to translate our in–vitro findings into the clinical reality.

  1. The paper is mainly descriptive and focused on its (not fully supported) conclusions.

    The new manuscript provides an important input on Breast Cancer Therapy Research via ASCs Exosomes. The manuscript is divided in a first section exploring EVs and Exosomes, a second part where authors highlight the current knowledge about Breast Cancer and a last part where the common points and possible in–vitro interactions between these fields are analyzed.

    Given that the interplay between ASCs Exosomes and TNBC is still unexplored and there is not enough literature about this topic, we propose a potential in–vitro technology to address TNBC cells. This should help and motivate not only our research group but even other authors to do research in this field.

Round 2

Reviewer 3 Report

Comments and Suggestions for Authors

Revising their manuscript, the authors have enhanced it by incorporating new figures and revising the text from the initial version. In this form, their paper could significantly contribute to the field.

Comments on the Quality of English Language

Minor editing of English language required

Reviewer 4 Report

Comments and Suggestions for Authors

In the revised article, the authors modified the manuscript and figures referred to the comments, and answered the questions comprehensively.